# Frontline Management of Epithelial Ovarian Cancer—Combining Clinical Expertise with Community Practice Collaboration and Cutting-Edge Research

**DOI:** 10.3390/jcm9092830

**Published:** 2020-09-01

**Authors:** Edward Wenge Wang, Christina Hsiao Wei, Sariah Liu, Stephen Jae-Jin Lee, Susan Shehayeb, Scott Glaser, Richard Li, Siamak Saadat, James Shen, Thanh Dellinger, Ernest Soyoung Han, Daphne Stewart, Sharon Wilczynski, Mihaela Cristea, Lorna Rodriguez-Rodriguez

**Affiliations:** 1Department of Medical Oncology and Therapeutics Research, City of Hope National Medical Center, Duarte, CA 91010, USA; sarliu@coh.org (S.L.); ssaadat@coh.org (S.S.); jashen@coh.org (J.S.); dapstewart@coh.org (D.S.); MCristea@coh.org (M.C.); 2Department of Pathology, City of Hope National Medical Center, Duarte, CA 91010, USA; cwei@coh.org (C.H.W.); swilczyn@coh.org (S.W.); 3Department of Surgical Oncology, City of Hope National Medical Center, Duarte, CA 91010, USA; stelee@coh.org (S.J.-J.L.); tdellinger@coh.org (T.D.); ehan@coh.org (E.S.H.); lorrodriguez@coh.org (L.R.-R.); 4Department of Population Sciences, City of Hope National Medical Center, Duarte, CA 91010, USA; sshehayeb@coh.org; 5Department of Radiation Oncology, City of Hope National Medical Center, Duarte, CA 91010, USA; sglaser@coh.org (S.G.); rli@coh.org (R.L.)

**Keywords:** epithelial ovarian cancer, frontline treatment, surgical debulking, adjuvant chemotherapy, maintenance therapy, PARP inhibitor, genetics counseling, clinical research, team medicine

## Abstract

Epithelial ovarian cancer (EOC) is the most common histology of ovarian cancer defined as epithelial cancer derived from the ovaries, fallopian tubes, or primary peritoneum. It is the fifth most common cause of cancer-related death in women in the United States. Because of a lack of effective screening and non-specific symptoms, EOC is typically diagnosed at an advanced stage (FIGO stage III or IV) and approximately one third of patients have malignant ascites at initial presentation. The treatment of ovarian cancer consists of a combination of cytoreductive surgery and systemic chemotherapy. Despite the advances with new cytotoxic and targeted therapies, the five-year survival rate for all-stage EOC in the United States is 48.6%. Delivery of up-to-date guideline care and multidisciplinary team efforts are important drivers of overall survival. In this paper, we review our frontline management of EOC that relies on a multi-disciplinary approach drawing on clinical expertise and collaboration combined with community practice and cutting edge clinical and translational research. By optimizing partnerships through team medicine and clinical research, we combine our cancer center clinical expertise, community practice partnership, and clinical and translational research to understand the biology of this deadly disease, advance therapy and connect our patients with the optimal treatment that offers the best possible outcomes.

## 1. Introduction

Epithelial ovarian cancer (EOC) is the most common histology of ovarian cancer, defined as epithelial cancer derived from the ovaries, fallopian tubes, or primary peritoneum [1]. It is the fifth most common cause of cancer-related death in women in the United States, with an estimated 21,750 new cases and 13,940 deaths in 2020 [2]. Because of a lack of effective screening [3] and non-specific symptoms, EOC is typically diagnosed at an advanced stage (FIGO stage III or IV) and approximately one third of patients have malignant ascites at initial presentation. The treatment of ovarian cancer is primarily limited to cytoreductive surgery and systemic chemotherapy. Despite the advances with new cytotoxic and targeted therapies, the five-year survival rate for all-stage EOC in the United States is 48.6% [4]. The delivery of up-to-date guideline care and multidisciplinary team efforts are important drivers of overall survival [5].

The City of Hope National Medical Center (COH) is an NCI-designated Comprehensive Cancer Center based in Duarte, California. Its service area includes Los Angeles, San Bernadino, Riverside, and Orange Counties. Together, these four counties are home to 46% of California’s total population. COH delivers high quality cancer care to this sizable demographic through its large network of community oncology practice clinics in the area. In this paper, we review the frontline management of EOC and how we combine our cancer center clinical expertise, community practice partnership, and clinical and translational research to understand the biology of this deadly disease and advance therapy.

## 2. Surgical Management

Cytoreductive surgery (debulking) plays a fundamental role in managing EOC. Studies show that survival is inversely correlated with the volume of residual disease after cytoreductive surgery [6,7,8,9,10,11,12,13]. Thus, the goal of surgery is to remove all visible disease [6,9,12,14,15,16,17,18]. In a 2011 meta-analysis of 11 retrospective studies of primary cytoreduction for advanced EOC, there was improved survival with optimal (residual disease ≤ 1 cm in maximum tumor diameter) versus suboptimal (residual disease > 1 cm in maximum tumor diameter) cytoreduction (hazard ratio (HR) 1.36, 95% CI 1.10–1.68), and further improved survival with no gross residual disease (HR 2.20, 95% CI 1.90–2.54) [19]. In a 2013 meta-analysis of 18 studies (retrospective and prospective) of women with stage IIB or higher EOC who underwent cytoreduction and platinum/taxane chemotherapy, each 10% increase in the proportion of patients undergoing complete cytoreduction was associated with a 2.3 month increase in median survival compared with a 1.8 month increase for optimal cytoreduction [14]. 

Furthermore, improved outcomes in advanced EOC have been shown in high volume hospitals (≥20 cases/year) and high-volume surgeons (≥10 cases/year) [20]. Given the importance of the extent of cytoreduction and volume of cases on outcome and the potential morbidity with an extensive major abdominal surgery, predicting which patients will be able to have at least an optimal cytoreduction is valuable. This is primarily performed through physical examination and computed tomography (CT) of the chest, abdomen, and pelvis. Diagnostic laparoscopy can also be utilized to help triage patients with primary debulking or neoadjuvant chemotherapy [21,22,23]. It is of utmost importance that a gynecologic oncologist experienced in extensive cytoreductive surgeries evaluates the patient to determine resectability, as achieving no gross residual disease or optimal cytoreduction largely depends on the judgment, experience, skill, and aggressiveness of the surgeon. Additionally, patient factors, such as age, performance status, medical comorbidities, and preoperative nutritional status, are important considerations, as some patients may not be able to tolerate an extensive cytoreduction. The commonly accepted criteria for unresectability include mesenteric root involvement, diffuse involvement of the stomach and/or large parts of the small or large bowel, extra-abdominal disease, infiltration of the duodenum and/or parts of the pancreas (not limited to the pancreatic tail), or involvement of the large vessels of the hepatoduodenal ligament, celiac trunk, or behind the porta hepatis [24].

Our strong partnership with community practices provides a large number of patients in Los Angeles and the Greater Los Angeles area with access to a high volume, high complexity cancer center. In addition to hysterectomy, bilateral salpingo-oophorectomy, and omentectomy, additional procedures can include small bowel resection, large bowel resection, stoma formation, diaphragm peritonectomy plus/minus segmental full-thickness diaphragm resection, splenectomy plus/minus distal pancreatectomy, segmental liver resection, cholecystectomy, partial stomach resection, and partial bladder/ureteral resection. We advocate against routine lymphadenectomy (pelvic, para-aortic) in patients undergoing cytoreduction for stage III or IV disease as it has not been shown to improve overall survival and results in increased postoperative morbidity [25]. However, we do resect suspicious or enlarged lymph nodes to achieve a complete or optimal cytoreduction. An intraperitoneal (IP) catheter for IP delivery of adjuvant chemotherapy may be placed in select patients who have obtained optimal primary cytoreduction, as combination treatment with intravenous (IV) and IP chemotherapy has been shown to prolong overall survival [26,27,28]; although newer trials have advocated for IV delivery of chemotherapeutics that may have similar outcomes but less morbidity than IP chemotherapy [29].

Patients referred to COH from our community clinics for the surgical management of EOC are assessed by our gynecologic surgical oncologist team and we perform primary cytoreduction for EOC in selected patients (those medically fit to undergo an extensive surgery and in whom it is deemed a resection to no gross residual disease or at least in whom an optimal debulking can be achieved) followed by adjuvant chemotherapy. Other patients deemed unresectable may undergo neoadjuvant chemotherapy and then re-evaluation for possible interval cytoreduction. We perform heated intraperitoneal chemotherapy (HIPEC) in a clinical trial setting for translational purpose toward personalized medicine. We collect biospecimens including peritoneal samples with and without tumor cells, blood samples before and after HIPEC. Paired tumor/normal whole exome sequencing (WES) and whole transcriptome sequencing (RNAseq) is performed for analyses of germline and somatic genomic landscapes, as well as gene expression phenotypes before and after treatment, including the assessment of driver mutations, mutation signatures, tumor mutation burden, and immune signatures. Hyperthermia increases the penetration of chemotherapy and increases the chemosensitivity of the cancer by impairing DNA repair. Additionally, hyperthermia induces apoptosis and activates heat-shock proteins that serve as receptors for natural killer cells, inhibits angiogenesis, and has a direct cytotoxic effect by promoting the denaturation of proteins. In a 2018 randomized trial, van Driel et al. reported a nearly 12-month survival benefit in those receiving HIPEC versus no HIPEC after undergoing at least an optimal interval cytoreduction with a similar rate of grade 3 or 4 adverse events between the two groups [30]. It is unclear if the IP administration, the heat, or the additional dose of chemotherapy is responsible for the benefit as all three interventions were utilized. These results are encouraging; however, further studies are needed before there is widespread adoption of this technique, which requires additional technical expertise [31,32].

Pressurized intraperitoneal aerosol chemotherapy (PIPAC) is another approach we are evaluating in the clinical trial setting. PIPAC is a novel minimally-invasive drug delivery system in which normothermic chemotherapy is administered into the abdominal cavity as an aerosol under pressure [33,34]. This approach uses the advantage of the physical properties of gas and pressure by generating an artificial pressure gradient and enhancing tissue uptake of the aerosolized chemotherapy. Due to high local bioavailability during PIPAC, lower concentrations of chemotherapy can be utilized, thus minimizing side effects and toxicity.

## 3. Gynecologic Pathology: Diagnostic Evaluation

Accurate pathologic diagnosis is the cornerstone of our treatment approach. When patients come to COH with a diagnosis of EOC made in the community, their surgical pathology is reviewed by our gynecologic pathology team. There are four major histologic types of ovarian epithelial tumors—serous, mucinous, endometrioid, and clear cell. High grade serous carcinoma (HGSC) is the most common, and lethal histologic subtypes of all ovarian epithelial malignancies are diagnosed, often presenting at an advanced stage. A subset of these patients carry germline mutations in double-strand DNA repair genes, such as BRCA1, BRCA2, RAD51c, and PALB2. Therefore, diagnosis of HGSC carries specific prognostic, therapeutic, and genetic implications. The ovarian cancer TCGA study showed that HGSC is characterized by a near universal p53 mutation [35]. Most of the p53 mutations lead to the overexpression or deletion of the protein, and these can be detected using immunohistochemistry. In morphologically ambiguous cases, performing a p53 mutation analysis may be helpful, and p53 mutation status can be used to temporally track patients’ tumors over time. Knowledge about the clinical and functional consequences of various p53 mutations is emerging. We perform whole-exome and RNA sequencing using the next generation sequencing platform for HGSC tumors. This allows us to define the p53 mutation profile in tumors and helps us to better understand clinical and treatment significance.

HGSC also displays genomic instability with high copy-number variations across the genome [36]. This unstable genomic landscape is a collective reflection of high tumor replication rate and the tumor cells’ underlying defective DNA repair mechanisms, specifically homologous recombination repair (HRR) [37]. In HGSC, which displays homologous recombination deficiency (HRD), tumors rely on alternative but error-prone pathways, including non-homologous end-joining and single-strand annealing repair pathways [38]. Women with germline BRCA1/2 mutations are enriched for the HRD phenotype [39]. The underlying HRD phenotype explains why some HGSC patients are sensitive to platinum-based chemotherapy (carboplatin, paclitaxel, or docetaxel) or poly-(ADP-ribose)-polymerase 1 (PARP1) inhibitors (such as olaparib and niraparib). Platinum-based chemotherapy induces synthetic lethality by covalent binding with DNA, forming DNA-platinum adducts that eventually trigger double strand break. PARP1 inhibitors impede the PARP1-mediated repair of DNA single strand breaks, a component of the HRR pathway. In HGSC with underlying HRD, double strand breaks cannot be repaired efficiently and their accumulation in the genome result in cell death [38].

HGSC is diagnosed using the MD Anderson histologic 2-tier system [40,41]. Corroborating with the molecular event of p53 mutation, the diagnosis of HGSC can be further supported by performing immunohistochemical staining for p53. HGSC is staged using the current American Joint Committee on Cancer/College of American Pathologists Cancer Staging Form and the FIGO Staging System. The molecular diagnosis of ovarian cancer subtypes that correlate with prognosis may also be adopted as standard procedure in the future. Verhaak et al. analyzed the TCGA database and revealed four ovarian tumor subtypes, each associated with a different prognosis [42]. 

## 4. Molecular Studies Available for Diagnostic or Therapeutic Decision Support and Emerging Translational Research

We perform extensive molecular testing, including whole exome sequencing, transcriptomic sequencing, copy number information, mismatch repair (MMR) deficiency, microsatellite instability (MSI) status, tumor mutation burden (TMB), HRD, and PD-L1 protein expression levels, using paired formalin-fixed paraffin-embedded tumor tissue and patient saliva or peripheral blood. This comprehensive approach allows us to detect somatic and germline mutations, clinically actionable mutations, potential therapeutic targets, and markers to help guide checkpoint inhibitor therapy. The genomic analysis makes tailored therapy possible and informs clinical trial options that best match with patient tumor genotype.

Germline and somatic BRCA1 and BRCA2 mutations are assessed in specific clinical contexts to inform genetic counseling and therapy selection. Younger age at presentation and family history of tubo-ovarian and breast cancer malignancies are risk factors suggestive of the presence of germline cancer predisposition syndrome. Referral to a genetic counselor and establishing germline mutation information is crucial for informing patients about BRCA-related cancer risks for themselves and their family members. Most importantly, this allows patients the opportunity to access BRCA-related cancer risk reduction surgeries (e.g., risk-reducing salpingo-ophorectomy, mastectomy), where the timing of surgery can be crucial to successful risk reduction.

Germline and somatic BRCA1/2 mutation information is also important for informing PARP inhibitor eligibility in Stage II, III, and IV HGSC patients post primary treatment. The NCCN guidelines recommend screening for BRCA mutations early in the treatment course to avoid the possibility of delay in instituting PARP inhibitor therapy [43].

HRD positivity is determined by BRCA mutation status (deleterious or suspected deleterious) or HRD/genomic instability score (mathematically derived from genomic assessment of loss of heterozygosity, telomeric allelic imbalance, and large-scale state transitions). Due to the inherent biocomputational complexity with HRD score derivation and inter-laboratory analytic variability, most large medical centers perform HRD testing on a research basis and not for routine clinical diagnostic use. 

Circulating miRNAs in blood and urine are being explored as potential early detection markers. However, the evidence on this approach is currently limited, and no consistent miRNA signatures have emerged [44,45,46]. The lack of reproducibility may be attributable, in part, to technical issues, such as different statistical modeling and approaches, the utilization of different miRNA detection platforms, and patient and tumor heterogeneity [46]. Besides early detection, liquid biopsy-based circulating tumor cells have been leveraged in a recent small pilot preclinical study to provide chemosensitivity information and therapy response prediction in patients presenting with recurrent ovarian cancer [47]. The quest for providing precision oncology to patients using minimally invasive liquid biopsies is expanding, and hopefully it will become a reality in the not so distant future.

With numerous genomic alterations present in HGSC, an integrative analytical approach is necessary to characterize the dominant biologic drivers of carcinogenesis, cancer progression, and prognosis. The TCGA (Cancer Genome Atlas Research Network) and CPTAC (Clinical Proteomic Tumor Analysis Consortium) investigators have paved the way for combining multiple omics in ovarian HGSC—including genomics, proteomics, and phosphoproteomics. Using transcriptomic data, TCGA has built a HGSC molecular taxonomy comprised of four subtypes: differentiated, mesenchymal, immunoreactive, and proliferative [35]. This framework was recapitulated using the proteomic data [36]. However, this molecular taxonomy does not correlate with patient survival [36]. Instead, proteomic signatures (cytoskeleton involved in invasion and migration, apoptosis, and epithelial junction/adhesion) showed more robust correlation with survival [36]. However, this proteomic signature is currently research-based only, awaiting further validation in larger independent cohorts, and is not currently used in clinical setting.

## 5. Adjuvant Chemotherapy

With the exception of patients with early-stage disease and low-grade cancers with a high cure rate, such as stage 1A and 1B grade 1 endometrioid ovarian cancer, mucinous carcinoma, and low grade serous carcinoma [48,49,50], patients with EOC who have undergone surgical debulking usually require adjuvant platinum- and taxane-based chemotherapy to reduce the risk of recurrence or prolong disease-free survival. Optimal time from surgery to initiate adjuvant chemotherapy has been shown to be 4–6 weeks [49,51]. Table 1 summarizes the main clinical studies of frontline treatment and maintenance of EOC. The standard adjuvant chemotherapy regimen includes: IV paclitaxel 175 mg/m^2^ and carboplatin AUC 5–6 every 3 weeks. Alternatively, dose dense weekly paclitaxel 80 mg/m^2^ and carboplatin AUC 5–6 every 3 weeks may be applied [52,53,54,55]—although this regimen has shown differing outcomes in different studies—the JGOG3016 study [52,56] showed a favorable outcome over every 3-week standard regimen, while the ICON-8 [55], and GOG-262 studies [53] failed to showed a significant improvement. The MITO-7 study used weekly paclitaxel 60 mg/m^2^ and carboplatin AUC 2 for up to 18 weeks—this regimen has a high tolerance and is effective for elderly patients or those with poor performance status [54]. Single agent carboplatin is also acceptable if patients cannot tolerate the combination treatment. Docetaxel is an acceptable taxane alternative to paclitaxel with equivalent efficacy [57]. Carboplatin plus liposomal doxorubicin is also an acceptable combination for adjuvant chemotherapy when patients cannot tolerate taxanes [58,59]. Recently, bevacizumab was incorporated into the adjuvant chemotherapeutic regimen, showing improved progression-free survival and also overall survival in the high risk of progression subgroup, including those with stage IV disease and inoperable or sub-optimally debulked stage III disease (ICON-7, GOG-218) [60,61], especially in patients with ascites [60,62,63]. 

In patients with EOC, the peritoneal cavity is usually the primary site of recurrence. Thus, the administration of adjuvant IV/IP chemotherapy to treat residual cancer cells with highly concentrated chemotherapeutics is an attractive approach. The GOG-172 study showed that IV paclitaxel 135 mg/m^2^ on day 1 plus IP cisplatin 75–100 mg/m^2^ on day 2 and IP paclitaxel 60 mg/m^2^ on day 8, every 3 weeks for up to six cycles, improved survival by 16 months in patients with optimally debulked stage III EOC compared with IV delivery of paclitaxel and cisplatin [27]; IP carboplatin is a suitable substitute for IP cisplatin in the GOG-252 study, as the median progression-free survival and overall survival were similar in the IP carboplatin and IP cisplatin arms [28]. However, the IV/IP chemotherapy regimen resulted in more side effects [64], including abdominal pain, catheter-related infection and blockage, and myelosuppression, all of which may delay treatment and compromise efficacy. We routinely use IV/IP adjuvant chemotherapy based on the favorable survival outcomes [27,65]. A recent publication showed that, when bevacizumab was added to IV/IV carboplatin and paclitaxel, IV/IP carboplatin and paclitaxel, or IV/IP cisplatin and paclitaxel, there was no significant difference in progression-free survival in all of these groups of patients [28]. Therefore, there is debate as to whether or not IP chemotherapy is still an acceptable option in primary adjuvant chemotherapy for patients with advanced EOC, given its higher toxicity, inconvenience, catheter complications, and uncertain long-term benefits [29]. At City of Hope, we have been treating patients with the IV/IP protocol. Due to recent advances in maintenance therapy, we are reconsidering if it is still necessary to perform the IP delivery of chemotherapeutics.

## 6. Maintenance Therapy

EOC patients who undergo surgical debulking and adjuvant chemotherapy still experience a high rate of disease recurrence. Thus, there is a need for effective maintenance therapy after adjuvant chemotherapy for patients with EOC to help prevent recurrence or prolong disease-free survival. In the past, patients who completed adjuvant chemotherapy usually underwent active surveillance with regular follow-up, labs, and imaging as needed. However, this practice was changed after the ICON-7 and GOG-218 studies showed clinical benefit by adding bevacizumab to the adjuvant chemotherapy regimen [59,60,61,62]. The ICON-7 study added bevacizumab (7.5 mg/kg) to IV paclitaxel and carboplatin on day 1, repeated every 3 weeks for 5–6 cycles, continuing bevacizumab for up to 12 additional cycles and showed a modest prolongation of progression-free survival by 2.4 months. Overall, survival was also increased in patients with a poor prognosis [61,66]. The GOG-218 study added bevacizumab to IV paclitaxel and carboplatin on day 1 of cycle 2 (15 mg/kg), every 3 weeks for up to 22 cycles. This regimen showed a significant benefit to progression-free survival (14.1 months vs. 10.3 months, *p* < 0.001). Patients with ascites who received bevacizumab in addition to paclitaxel and carboplatin had significantly improved progression-free survival and overall survival compared to those who received paclitaxel and carboplatin alone [63]. However, maintenance with PARP inhibitors may be favored over bevacizumab due to improved survival.

Following success in treating recurrent EOC, PARP inhibitors have also recently become an attractive choice for maintenance after adjuvant chemotherapy in newly diagnosed EOC patients. Olaparib was FDA-approved (2018) for the maintenance treatment of adult patients with deleterious or suspected deleterious germline or somatic BRCA-mutated advanced EOC who are experiencing a complete or partial response to first-line platinum-based chemotherapy. This is based on the SOLO-1 study [67], a randomized, double-blind, placebo-controlled, multi-center trial that compared the efficacy of olaparib with placebo in patients with BRCA-mutated advanced ovarian, fallopian tube, or primary peritoneal cancer following first-line platinum-based chemotherapy. After a median follow-up of 41 months, the risk of disease progression or death was 70% lower with olaparib than with placebo. In May 2020, the FDA expanded the indication of olaparib to include its combination with bevacizumab for first-line maintenance treatment of adult patients with advanced EOC who have complete or partial response to first-line platinum-based chemotherapy and whose cancers are HRD-positive, defined by either a deleterious or suspected deleterious BRCA mutation and/or genomic instability score. This recommendation was based on the study by Ray-Coquard et al. [68], which showed that, in patients with advanced EOC receiving first-line standard therapy bevacizumab, the addition of maintenance olaparib provided a significant progression-free survival benefit, which was substantial in patients with HRD-positive tumors (37.2 vs. 17.7 months). Patients with HRD-positivity but without a BRCA mutation also had significantly improved progression-free survival (28.1 vs. 16.6 months).

Niraparib, another PARP inhibitor, was granted approval by the FDA in April 2020 as a first-line maintenance treatment of adult patients with advanced EOC who experienced a complete or partial response to first-line platinum-based chemotherapy, regardless of biomarker status. This recommendation is based on the PRIMA study [69] (Table 1) which showed that patients with newly diagnosed advanced EOC who had a response to platinum-based chemotherapy and received niraparib had significantly longer progression-free survival than those who received placebo (13.8 vs. 8.2 months), regardless of the presence or absence of HRD. We use niraparib for patients without BRCA mutation or HRD, or patients with unknown BCRA/HRD status.

Additional maintenance options are being studied in clinical trials, including new PARP inhibitors, anti-angiogenesis agents, immune checkpoint inhibitors, agents targeting other signal transduction pathways, and new rational combinations. We expect to have improved maintenance options in the future to further reduce recurrence and prolong disease-free survival. Choosing the right maintenance therapy for each patient is highly complex and benefits from multi-disciplinary discussion. At COH, the community oncologists have access to the COH Gynecologic Cancer Tumor Board (discussed further below) to present their challenging cases for in-depth discussion.

## 7. Genetic Counseling

HGSC is a single case indicator for germline genetic testing [70]. Germline genetic testing should be considered both due to the relatively high percentage of hereditary ovarian cancer with some studies estimating that more than 20% is hereditary in etiology [71,72,73], and due to the potential for treatment implications [74]. Generally, it is preferable for an individual to undergo germline testing as soon as diagnosis occurs [75,76]. This allows ample time to obtain and disclose results, especially in the setting of a patient who may have a guarded prognosis. Urgent testing of BRCA1/2 and other breast cancer genes with high or moderate penetrance by multi-gene panel can currently be performed. While this strategy is often used for women with breast cancer undergoing surgical decision-making, it can also be employed in the gynecologic oncology setting to provide results that may affect eligibility for PARP inhibitors or other therapies in a timely manner.

Germline testing in an affected individual is the most informative strategy and can help clarify risk for relatives. Close female relatives may have increased empiric risk to develop EOC, although older studies may include some families with risk alleles that would be identified by current technology [77,78]. The ascertainment of a multi-generational pedigree allows both for appropriate test selection as well as for proper assessment of family structure and identification of at-risk relatives [79]. Pedigree assessment in the setting of genetic counseling can also facilitate understanding of social relationships between relatives to help develop appropriate strategies to encourage familial communication about risk.

Germline testing for women with EOC at our center typically includes evaluation via a multi-gene panel to include EOC risk genes beyond BRCA1/2, such as the mismatch repair (Lynch syndrome) genes, BRIP1, RAD51C, and RAD51D [71,80]. Beyond informing therapeutic strategy, germline testing in the setting of appropriate counseling can have significant implications for patients and family members. Germline testing can help stratify the risk of developing other cancers and guide the development of appropriate management strategies, especially as the prognosis for EOC improves with better treatment options. For example, patients with Lynch syndrome are at significantly elevated risk to develop colorectal cancer [81] and patients with pathogenic alterations in the BRCA genes are at significantly elevated risk to develop breast cancer [82]. Understanding a patient’s risk may help prevent a second primary cancer, especially in the setting of well-controlled ovarian disease or in the setting where the development of a new cancer may interfere with the patient’s current treatment.

Germline testing may be even more impactful in terms of implications for relatives. Identifying an ovarian cancer risk allele can allow relatives with the same allele to undergo preventative measures, such as risk-reducing salpingo-oopherectomy, which is especially relevant when screening is not effective. Moreover, in some cases, over-treatment may be avoided in relatives who do not carry the risk allele but who may have otherwise chosen to move forward with preventative measures or screening due to concerns over risk, based on family history. Many genes implicated in EOC in the setting of a monoallelic pathogenic variant also have implications for typically childhood-onset syndromes in the setting of biallelic pathogenic variants. For example, biallelic variants in BRCA1/2, BRIP1, and RAD51C [83,84,85,86] are associated with Fanconi anemia and biallelic variants in the mismatch repair genes are associated with Constitutional Mismatch Repair Deficiency syndrome [81]. Thus, individuals contemplating childbearing may also wish to learn their germline status to inform reproductive decisions.

Importantly, negative somatic testing does not obviate the need for germline testing. Reasons for this can include the loss of a germline mutation in the tumor, limited analysis of the tumor genome, and differences in variant calling between somatic and germline laboratories. Conversely, somatic testing may identify variants that are germline in origin [87,88]. Therefore, patients should be counseled about this possibility, and if somatic results are available, they should be reviewed to help inform germline test selection. Other genes may also be included based on clinical suspicion and the evaluation of additional personal and family history. Reevaluation should be considered over time as changes to the family history, as well as advances in the field of cancer genetics, occur [79].

## 8. Team Medicine: Optimizing Partnerships and Clinical Research

We have a number of initiatives to ensure the inclusion of our community partners in research, education, and the integration of research-based advances into novel therapeutics by clinical trials. We aim to personalize therapy for patients so our community physicians can recommend improved therapy considerations, including clinical trials beyond the standard of care. One way we achieve this is via comprehensive molecular testing. All EOC patients at COH undergo GEM ExTra^®^ testing (facilitated by TGen, a COH affiliate). This test reports clinically actionable mutations, copy number alterations, transcript variants, and fusions, detected in any gene in patient DNA or RNA. The goal is to uncover true tumor-specific (somatic) alterations by comparing the sequence of the tumor against the paired normal DNA from each patient. The test also includes whole-transcriptome RNA profiling, interrogating the patient’s tumor transcriptome for fusions and transcriptional variants known to be relevant to cancer (e.g., EGFR vIII). Each tumor’s cancer-specific alterations are then queried against a proprietary knowledge base algorithm to identify potential therapeutic associations. The final report provides the physician with a list of FDA-approved agents that are associated with tumor-specific DNA alterations, as well as biomarker summaries on the variants found and tumor-specific evidence for drug matches, including matches with investigational agents, as available on clinicaltrials.gov. The results are reviewed by our multidisciplinary gynecologic cancer research team to aid in treatment decision-making, highlight on-going studies and identify study candidates.

Our current clinical research portfolio in the frontline management of EOC focuses on developing superior treatment options for patients that reduce recurrence and prolong disease-free survival. We are exploring the use of HIPEC and PIPAC in the clinical trial setting as well as novel drug combinations that help to tailor and personalize treatment for superior results. Our HIPEC trial includes studying the molecular changes triggered by HIPEC to identify molecular signatures of response. Our PIPAC trial is the first in the United States to study aerosolized, pressurized chemotherapy for patients with peritoneal carcinomatosis, including ovarian cancer. Our community oncologists play an important role in these studies by referring patients, thereby allowing us to complete accrual expeditiously.

## 9. Summary

Management of EOC requires a multi-disciplinary approach, drawing on clinical expertise and collaboration combined with community practice and cutting edge clinical and translational research. Our goal is to understand the biology of this disease, advance therapy and connect our patients with the optimal treatment that offers the best possible outcomes.

## Figures and Tables

**Table 1 jcm-09-02830-t001:** Major clinical trials on frontline treatment of epithelial ovarian cancer.

Study	Patients	Experimental	Control	Progression Free Survival	Overall Survival
**JGOG 3016 [52,56]**	Stage II-IV EOC	three-weekly carboplatin (AUC 6) and weekly paclitaxel (80 mg/m^2^) for six cycles	three-weekly carboplatin (AUC 6) and paclitaxel (180 mg/m^2^) for six cycles	28.0 vs. 17.2 months; HR 0.71, 95% CI 0.58–0.88; *p* = 0.0015	100.5 vs. 62.2 months (HR 0.79, 95% CI 0.63–0.99; *p* = 0.039
**MITO-7 [54]**	FIGO stage IC-IV EOC	Weekly carboplatin (AUC 2) and paclitaxel (60 mg/m^2^) for 18 weeks	three-weekly carboplatin (AUC 6) and paclitaxel (175 mg/m^2^) for six cycles	18.3 vs. 17.3 months; HR 0·96, 95% CI 0·80–1.16; *p* = 0·66	-
**ICON-8 [55]**	FIGO stage IC-IV EOC	Group 2: three-weekly carboplatin (AUC 5/6) and weekly paclitaxel (80 mg/m^2^) for six cycles Group 3: Weekly carboplatin (AUC 6) and paclitaxel (60 mg/m^2^) for 18 weeks	Group 1: three-weekly carboplatin (AUC 5/6) and paclitaxel (175 mg/m^2^) for six cycles	Group 1 vs. Group 2 vs. Group 3: 17.7 vs. 20.8 vs. 21.0 Group 2 vs. Group 1: *p* = 0.35 Group 3 vs. Group 1: *p* = 0.51	-
**GOG-172 [27,65]**	FIGO stage III with optimal debulking	paclitaxel 135 mg/m^2^ continuous iv infusion over 24 h on day 1, cisplatin 100 mg/m^2^ IP on day 2, paclitaxel 60 mg/m^2^ IP on day 8 for six cycles	paclitaxel 135 mg/m^2^ continuous IV infusion over 24 h on day 1, cisplatin 75 mg/m^2^ IV on day 2 for six cycles	23.8 vs. 18.3 months; HR 0.80, 95% CI 0.64–1.00; *p* = 0.05	65.6 vs. 49.7 months; HR 0.75, 95% CI, 0.58–0.97; *p* = 0.03 61.8 vs. 51.4 months; Adjusted HR 0.77; 95% CI, 0.65–0.90; *p* = 0.002
**GOG-252 [28]**	FIGO stage II-IV EOC	paclitaxel 80 mg/m^2^ IV on days 1, 8, and 15 plus carboplatin AUC 6 IP on day 1 every 21 days for cycles 1–6 plus bevacizumab 15 mg/kg IV every 21 days for cycles 2–22 paclitaxel 135 mg/m^2^ IV on day 1 plus cisplatin 75 mg/m^2^ IP on day 2 plus paclitaxel 60 mg/m^2^ IV on day 8 every 21 days for cycles 1–6 plus bevacizumab 15 mg/kg IV every 21 days for cycles 2–22	paclitaxel 80 mg/m^2^ IV on days 1, 8, and 15 plus carboplatin AUC 6 IV on day 1 every 21 days for cycles 1–6 plus bevacizumab 15 mg/kg IV every 21 days for cycles 2–22	IV vs. IP-carboplatin vs. IP-cisplatin: 24.9 vs. 27.4 vs. 26.2 months IP-carboplatin: HR 0.93, 95% CI 0.80–1.07 IP-cisplatin: HR 0.98, 95% CI 0.84–1.13	IV vs. IP-carboplatin vs. IP-cisplatin: 75.5 vs. 78.9 vs. 72.9 months IP-carboplatin: HR 0.95, 95% CI 0.80–1.13 IP-cisplatin: HR 1.05, 95% CI; 0.88–1.24;
**GOG-262 [53]**	FIGO stage III-IV EOC	three-weekly carboplatin (AUC 6) and weekly paclitaxel (80 mg/m^2^), plus/minus three-weekly bevacizumab 15 mg/kg for six cycles	three-weekly carboplatin (AUC 6) and paclitaxel (175 mg/m^2^), plus/minus three-weekly bevacizumab 15 mg/kg for six cycles	With bevacizumab: 14.9 vs. 14.7 months; HR 0.99, 95% CI 0.83–1.20; *p* = 0.60 Without bevacizumab: 14.2 vs. 10.3 months; HR 0.62, 95% CI 0.40–0.95; *p* = 0.03	With and without bevacizumab: 40.2 vs. 39.0 months; HR 0.94; 95% CI, 0.72–1.2
**SOLO-1 [67]**	FIGO stage III or IV high-grade serous or endometrioid EOC patients with a deleterious or suspected deleterious germline or somatic BRCA1/2 mutation, completed frontline platinum-based chemotherapy	olaparib	placebo	Not reached vs. 13.8 months; HR 0.30, 95% CI 0.23–0.41); *p* < 0.0001 3-year: 60% vs. 27%; 4-year: 53% vs. 11%	-
**PAOLA-1 [68]**	FIGO stage III or IV high-grade EOC patients after first-line treatment with platinum–taxane chemotherapy plus bevacizumab	olaparib plus bevacizumab	placebo plus bevacizumab	22.1 vs. 16.6 months; HR 0.59; 95% CI 0.49–0.72; *p* < 0.001 HRD plus BRCA mutation: 37.2 vs. 17.7 months; HR 0.33, 95% CI 0.25–0.45 HRD minus BRCA mutation: 28.1 vs. 16.6 months; HR 0.43, 95% CI 0.28–0.66	-
**PRIMA [69]**	FIGO stage III or IV high-grade serous or endometrioid EOC patients after first-line treatment with platinum-based chemotherapy	niraparib	placebo	Overall: 13.8 vs. 8.2 months; HR .62, 95% CI 0.50–0.76; *p* < 0.001 HRD-positive: 21.9 vs. 10.4 months; HR 0.43, 95% CI 0.31–0.59; *p* < 0.001	-

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
