# Peer review of "Frontline Management of Epithelial Ovarian Cancer—Combining Clinical Expertise with Community Practice Collaboration and Cutting-Edge Research"

_jcm, 2020, doi:10.3390/jcm9092830_

Round 1
Reviewer 1 Report
The manuscript entitled “Frontline management of epithelial ovarian cancer – combining clinical expertise with community practise collaboration and cutting-edge research” is a technically sound and very well thought the manuscript. This work summarises the newest findings in the management of EOC care. Before the manuscript is accepted for publication, there are minor comments and suggestion authors need to address. Authors should put more effort into the ‘Molecular studies available for diagnostic or therapeutic decision support”, and should try to present a broader perspective of the molecular studies including also metabolomic and proteomic, as well as the possibility of using circulating miRNAs for early diagnosis of EOC. I think some chapters need to add a table as a summary. If possible, this would make reading the manuscript easier.
Author Response
The manuscript entitled “Frontline management of epithelial ovarian cancer – combining clinical expertise with community practise collaboration and cutting-edge research” is a technically sound and very well thought the manuscript. This work summarises the newest findings in the management of EOC care. Before the manuscript is accepted for publication, there are minor comments and suggestion authors need to address. Authors should put more effort into the ‘Molecular studies available for diagnostic or therapeutic decision support”, and should try to present a broader perspective of the molecular studies including also metabolomic and proteomic, as well as the possibility of using circulating miRNAs for early diagnosis of EOC. I think some chapters need to add a table as a summary. If possible, this would make reading the manuscript easier.
We thank the reviewer for the positive comments. We added more paragraphs in the section entitled: “Molecular studies available for diagnostic or therapeutic decision support, and emerging translational research” discussing the molecular studies in EOC in details.
Reviewer 2 Report
Comments to JCM-891025
Frontline management of epithelial ovarian cancer - combining clinical expertise with community practice collaboration and cutting-edge research.
This review is elegant even if not comprehensive.
Minor changes are suggested.
Page 6, line 119: please precise inclusion/exclusion criteria of your HIPEC trial.
Page 7, line 131: please precise inclusion/exclusion criteria of your HIPEC trial.
Page 11, line 224: Please, also report other studies using liquid biopsies (i.e. A pilot study of the predictive potential of chemosensitivity and gene expression assays using circulating tumour cells from patients with recurrent ovarian cancer. Guadagni S., et al. Int. J. Mol. Sci. 2020, 21(13), 4813. https://doi.org/10.3390/ijms21134813).
I suggest minor changes in the style of 2 Sections: 1) Team Medicine: Optimizing Partnerships; 2) Clinical Research. In some parts there is an excessive self-referentiality (almost publicity) of the City of Hope National Medical Center (COH). Please, consider that this review article is addressed to the whole word (including nations with social, political and economic situations apart from the USA).
In particular, please, at page 22, restructure the text from line 435 to line 438 (too celebrative of COH community oncologists).
Author Response
This review is elegant even if not comprehensive.
Minor changes are suggested.
Page 6, line 119: please precise inclusion/exclusion criteria of your HIPEC trial.Page 7, line 131: please precise inclusion/exclusion criteria of your HIPEC trial.
We thank the reviewer for the positive comments. The I/E criteria is not much different from other published HIPEC trials, but we collect blood/tissue before and after HIPEC for molecular studies toward personalized medicine. Added in the text.
Page 11, line 224: Please, also report other studies using liquid biopsies (i.e. A pilot study of the predictive potential of chemosensitivity and gene expression assays using circulating tumour cells from patients with recurrent ovarian cancer. Guadagni S., et al. Int. J. Mol. Sci. 2020, 21(13), 4813. https://doi.org/10.3390/ijms21134813).
We added more paragraphs in the Molecular studies section, which include discussion of liquid biopsy, added reference recommended.
I suggest minor changes in the style of 2 Sections: 1) Team Medicine: Optimizing Partnerships; 2) Clinical Research. In some parts there is an excessive self-referentiality (almost publicity) of the City of Hope National Medical Center (COH). Please, consider that this review article is addressed to the whole word (including nations with social, political and economic situations apart from the USA). In particular, please, at page 22, restructure the text from line 435 to line 438 (too celebrative of COH community oncologists).
We have adjusted the text in the Team Medicine: Optimizing Partnerships and Clinical Research section, combining the two sections and editing to address the reviewers comments.
Reviewer 3 Report
Kudos to the authors for their well written manuscript detailing their approach to an unmet need and that is optimization of care for all women with ovarian cancer. I have a few suggestions/edits to hopefully strengthen the papers focus in some areas and scope in others. I would change the flow of the paper so that it mirror the clinical flow of patients in COH (incidence, referral, genetic counseling, suitability for surgery, adjuvant therapy recs, maintenance therapy, etc….)
Line |
Comment |
21, 38 |
I would use the intro to define the problem which is not necessarily the most common histology, but rather the severity and frequency of the diagnosis of EOC |
26, 44 |
I would rephrase instead of saying limited to, to say initial treatment consists of a combination of surgery and chemotherapy |
46 |
I suspect 47.6 could be rounded up without losing meaning |
53 |
I think this line does a good job of stating the focus of the paper |
68 |
I would define what optimal versus complete is as this can be confusing. Better terminology may be no gross residual disease |
70 |
Need space between % and increase |
|
You mention two different meta-analyses. I suspect there is overlap in studies reported on. Maybe best approach is to condense and comment on most relevant, contemporary one I think a comment on the natural history contribution that may impact the resectability of the disease is warranted as that contributes to role of neoadjuvant chemotherapy. I would look at: Horowitz et al. J clin Oncol 2015 Mar 10;33(8): 937-43 |
78 |
Agree |
93 |
There are actually three GOG trials comparing IV to IP/IV |
93 |
Might want to reference these newer trials with that statement regarding IV versus IP |
97 |
I think this high volume discussion actually dovetails in better with line 78 line of thought with the experienced gyn onc |
107,114 |
There is an SGO white paper regarding who should be considered and referenced Wright et al. Gyn onc. Vol 143 Issue 1 p. 3-15. October 1, 2016 |
125 |
There are some limitations as to the applicability to general practice of the van Driel et al hyperthermia paper that should be referenced, also as to explain why with the referenced improved outcomes still mean HIPEC is still only in clinical trial setting |
131 |
Sounds exciting |
140 |
This section seems a little disjointed as it discusses histologic diagnosis and outcomes with risk reducing surgery for BRCA carriers |
160 |
I think this is a good explanation of HRD and its implications. Would this also be a good area to discuss how HRD tumors are particularly more susceptible to PARP inhibition? |
217 |
Early detection seems like it should be its own separate section. Is this relevant to the community based/expert perspective that the paper is espousing? |
275 |
There should be a line about how you at COH tailor therapy, decide on which regimens, because you’re right,there are multiple approved options |
293 |
Can you comment on the clinical uptake of Avastin in the community and the risk/benefit decision analysis that comes in to play regarding the 3 month improvement in PFS with Avastin, especially as it related to the recent emergence of PARP inhibition |
303 |
Can the authors comment on when they’d combine avastin with olaparib versus PARP alone (niraparib) for first line maintenance therapy |
332 |
This section seems out of place considering genetic and mutational testing (germline included) earlier I think the section is well written |
392 |
I like this approach, but I think I would like more detail as to when individualized treatment is utilized (recurrent plat resistant?) b/c I doubt any mutational analysis would trump standard of care established by randomized controlled trial for frontline treatment; What about educational efforts for your community partners? |
